# Quality of Life in Systemic Lupus Erythematosus and Other Chronic Diseases: Highlighting the Amplified Impact of Depressive Episodes

**DOI:** 10.3390/healthcare12020233

**Published:** 2024-01-17

**Authors:** Diego Primavera, Mauro Giovanni Carta, Ferdinando Romano, Federica Sancassiani, Elisabetta Chessa, Alberto Floris, Giulia Cossu, Antonio Egidio Nardi, Matteo Piga, Alberto Cauli

**Affiliations:** 1Department of Medical Sciences and Public Health, University of Cagliari, 09127 Cagliari, Italy; maurogcarta@gmail.com (M.G.C.); federicasancassiani@yahoo.it (F.S.); elis.chessa@gmail.com (E.C.); giuliaci@hotmail.com (G.C.); matteo.piga@unica.it (M.P.); alberto.cauli@unica.it (A.C.); 2Department of Public Health and Infectious Diseases, University of Rome La Sapienza, 00100 Rome, Italy; ferdinando.romano@uniroma1.it; 3UOC Reumatologia, Azienda Ospedaliero Universitaria Cagliari, 09124 Cagliari, Italy; 4Panic and Respiration Laboratory, Federal University of Rio de Janeiro, Rio de Janeiro 22725, Brazil; antonioenardi@gmail.com

**Keywords:** SLE, depression, quality of life

## Abstract

Background: Extensive research has explored SLE’s impact on health-related quality of life (H-QoL), especially its connection with mental wellbeing. Recent evidence indicates that depressive syndromes significantly affect H-QoL in SLE. This study aims to quantify SLE’s impact on H-QoL, accounting for comorbid depressive episodes through case-control studies. Methods: A case-control study was conducted with SLE patients (meeting the ACR/EULAR 2019 criteria of age ≥ 18). The control group was chosen from a community database. H-QoL was measured with the SF-12 questionnaire, and PHQ-9 was used to assess depressive episodes. Results: SLE significantly worsened H-QoL with an attributable burden of 5.37 ± 4.46. When compared to other chronic diseases, only multiple sclerosis had a worse impact on H-QoL. Major depressive episodes had a significant impact on SLE patients’ H-QoL, with an attributable burden of 9.43 ± 5.10, similar to its impact on solid cancers but greater than its impact on other diseases. Conclusions: SLE has a comparable impact on QoL to serious chronic disorders. Concomitant depressive episodes notably worsened SLE patients’ QoL, exceeding other conditions, similar to solid tumors. This underscores the significance of addressing mood disorders in SLE patients. Given the influence of mood disorders on SLE outcomes, early identification and treatment are crucial.

## 1. Introduction

Systemic lupus erythematosus (SLE) is a persistent autoimmune disorder that predominantly manifests in individuals within the youthful and middle-age demographics. It exhibits a marked predilection for females over males, with a gender ratio of 10:1. The annual incidence of SLE ranges from 0.3 to 31.5 cases per 100,000 individuals, and the adjusted prevalence approaches or exceeds 50 to 100 cases per 100,000 individuals [1]. The clinical spectrum of SLE is remarkably diverse, encompassing a range of manifestations from mild cutaneous presentations to severe outcomes such as catastrophic organ failure and complications related to obstetrics [2].

Among the organs most significantly impacted by SLE, the kidneys and skin stand out, exhibiting heightened susceptibility to pathological alterations [3]. Despite advancements in therapeutic options and improved survival rates, SLE remains an incurable disease [4]. Characterized by immune dysregulation and abnormal autoantibody production [5], SLE manifests as a prolonged condition with diverse clinical symptoms and multi-organ involvement.

Upon diagnosis, patients face the challenge of managing the disease over an extended period, with long-term medication and recurrent flare-ups imposing a considerable mental and economic burden. This significantly impacts their quality of life (QoL), work, and education. When evaluating the therapeutic efficacy of a disease, it is imperative to consider not only the biological indicators for physical function but also the psychological and social aspects to comprehensively assess overall function, i.e., QoL.

As health needs have evolved, the medical model has shifted towards a biological–social–psychological paradigm, emphasizing the significance of QoL in the medical field. In this context, QoL, a complex concept interpreted differently across disciplines, refers to an individual’s perception of their living conditions based on existing values and cultural systems, intertwined with their expectations and living standards [6]. This holistic perspective underscores the importance of considering various dimensions when assessing the impact of SLE on individuals’ lives.

The topic of compromise in health-related quality of life (H-Qol) associated with SLE has been explored in depth by hundreds of studies over the last 30 years and some review and meta-analysis works have taken stock of the current state of research [7,8]. 

The literature has examined the comparative aspects to other rheumatologic diseases such as rheumatoid arthritis; it becomes evident that, in the case of SLE, the psychological components are more compromised, whereas in rheumatoid arthritis, those related to physical components are more affected Despite these findings, the relationship between mental well-being and H-Qol in SLE has not been sufficiently investigated.

In a meta-analysis of studies employing the SF-36 as a measure of QoL, it was observed that the extent of organ damage exhibited a significantly stronger negative correlation with the SF domain of physical functioning (*p* < 0.001) while showing a less pronounced negative correlation with the SF-36 domain of mental health (*p* = 0.268) [9]. This outcome is seemingly counterintuitive, due to several noteworthy factors. First, the previous research has established an association between SLE and anxiety as well as between SLE and mood disorders, supported by evidence pointing to a bidirectional interaction between inflammatory pathways and anxiety/mood disorders linked to SLE [10]. Second, given the clinical manifestations of SLE including cerebral involvement, it can exert profound effects on mental well-being [11]. Third, it is widely recognized that medications proven effective in SLE therapy can potentially induce side effects affecting mental health [12]. Lastly, the effect of the stress of living with a chronic illness must be considered. It is only recently that emerging evidence has begun to suggest that the presence of depressive syndromes may serve as a determinant for a significant impairment of H-QoL among SLE patients [13,14].

The primary aim of this study was to assess the extent to which SLE influences H-QoL. To achieve this objective, we employed a comparative approach, drawing on findings from analogous case-control studies that utilized the same database for control group selection. This allowed us to ensure reliable comparison measures. As an additional objective, we sought to conduct an investigation to quantify the impact of comorbid depressive episodes on exacerbating the deterioration of QoL.

## 2. Materials and Methods

### 2.1. Design

Case-control study.

### 2.2. Study Sample

SLE patients at the Lupus Clinic of Cagliari were enrolled in a cross-sectional study between April 2019 and February 2020. Study inclusion criteria were: (a) fulfillment of the ACR/EULAR 2019 criteria, (b) being of age ≥ 18 years old, and (c) being capable of giving consent. 

Demographics, serological data, clinical data, and ongoing medications including the prednisone (PDN) equivalent daily dose, were recorded. Disease activity was assessed by the Systemic Lupus Erythematosus Disease Activity Index 2000 (SLEDAI-2K) [15] and the Physician Global Assessment (PGA) [16]. Organ damage was measured according to the SLICC/ACR Damage Index (SDI) [17].

The study sample included patients consecutively admitted at the Rheumatology Unit of the University Clinic and AOU of Cagliari, Cagliari, Italy.

The control group was drawn from a community database [18]. For each case, a matched age (same age) and sex-healthy control cell was created. From each cell, four controls were randomly selected. Once a drawn-out subject was included in a block, they were automatically excluded from the remaining blocks. 

Patients with SLE were compared with the attributable burden on QoL due to other chronic diseases (multiple sclerosis, major depressive disorder, Wilson’ disease, carotid atherosclerosis, solid cancer, PTSD, celiac disease, obsessive compulsive disorder, and specific phobia).

Recall bias and selection bias are minimized through the careful study design and the participant recruitment procedures.

First, regarding the recall bias, the information gathered in the study involves objective and concrete data, such as demographics, serological and clinical data, ongoing medications, disease activity, and organ damage. These parameters are less reliant on the participants’ memory or subjective interpretation, reducing the likelihood of recall bias. Additionally, the study’s time frame (between April 2019 and February 2020) is relatively short-term, further mitigating the potential for participants to inaccurately remember details over extended periods.

Second, with respect to selection bias, the inclusion criteria for SLE patients and controls were explicitly defined. SLE patients were enrolled based on specific criteria, including the fulfillment of the ACR/EULAR 2019 criteria, being of an age ≥ 18 years old, and the capability of giving consent. This clear and stringent inclusion criteria process helped ensure that the participants selected for the study are relevant to the research objectives, minimizing the potential for selection bias.

Moreover, the study sample includes patients consecutively admitted to the Rheumatology Unit at the University Clinic and AOU of Cagliari, which adds to the representativeness of the sample. The control group, drawn from a community database, underwent a careful matching process, creating matched age and sex-healthy control cells to enhance comparability. The random selection of controls from each cell further contributes to minimizing selection bias.

### 2.3. Tools and Assessment

Ad hoc forms to collect demographics, serological and clinical data, and ongoing medications including the prednisone (PDN) equivalent daily dose, were recorded. 

Among the various instruments designed to assess QoL, the SF-36 questionnaire encompasses 36 questions addressing domains such as ‘physical and social functioning,’ ‘role physical and role emotional,’ ‘general and mental health,’ and ‘bodily pain and energy’ [19]. In an effort to streamline the assessment procedures and address concerns about the extended administration time associated with the SF-36, an evaluation of the SF-12 questionnaire was undertaken. The SF-12, a self-report questionnaire (Short Form Health Survey—12 items), mirrors the dimensions of the SF-36 but comprises a reduced set of 12 questions, thereby facilitating a more expeditious administration. Comparative investigations have demonstrated that both the SF-36 and SF-12 exhibit similar psychometric characteristics and yield comparable results [18,20]. In the present study, the Italian version of the SF-12 with a Cronbach’s alpha of 0.7 [21] was employed to assess H-QoL. The SF-12 measures physical functioning, role limitations due to physical health problems, bodily pain, general health, vitality (energy/fatigue), social functioning, role limitations due to emotional problems, and mental health (psychological distress and psychological well-being). Two composite scores—the physical component score (PCS) and the mental component score (MCS)–are computed from all 12 items using a standard scoring algorithm. [22]. The PCS score primarily focuses on physical functioning, role-physical functioning, bodily pain, general health, and vitality scales. The MCS focuses on vitality, social functioning, role emotional functioning, and emotional well-being scales. Both the PCS and MCS scores range from 0 to 100; a higher score indicates a better health status. [23]. The total score (range: 12–47) is the result of the answers given by the Likert scale; a higher score indicates a better perceived H-QoL.

The Patient Health Questionnaire 9 Items (PHQ-9) is a self-administered screening questionnaire meant to identify depressive episodes [24]. It detects, in the form of specific questions, the presence of all the nine DSM core criteria for the diagnosis of a major depressive episode; the score for each item ranges from zero (complete absence) to three (almost every day). The score resulting from the sum of the answers for each item identifies if >8 indicates mild to severe depression [25].

### 2.4. Statistical Analyses

Data were collected anonymously using unique subject ID numbers, and the information was entered into a dedicated database. The measurable impact of SLE on the degradation of H-QoL was calculated as the difference between the SF-12 score of a control group matched for sex and age within the community and that of the study cohort. A one-way analysis of variance (ANOVA) was employed to assess differences in H-QoL and Patient Health Questionnaire-9 (PHQ9) scores among the analyzed subgroups. Furthermore, the burden of SLE in exacerbating H-QoL was compared to the burden associated with other diseases, as determined in previous case-control studies that used the same dataset to select the control groups [26,27,28,29,30,31,32]. Subsequently, the burden directly linked to depression was calculated as the average deviation in the SF-12 scores between individuals with SLE without depression and those with solid cancer and depression. A similar approach was also used in prior case-control studies that utilized the same dataset to form control groups. Thus, we compared the burden attributed to experiencing a major depressive episode with that observed in the presence of a major depressive episode in other chronic conditions.

## 3. Results

The demographic data of the study sample with SLE and of the control sample of people without SLE shows (men ± standard deviation of age and number and percentage of females) two samples perfectly balanced due to randomization by the blocks method. The mean score on the SF-12 scale was 32.96 ± 7.09 in SLE cases versus 38.33 ± 6.04 in controls; the difference was statistically significant (ANOVA 1 way DF 1, 53, 54; F = 18.161, *p* < 0.0001). Starting from these data, it can be calculated that the attributable burden to the SLE in worsening the H-Qol was 5.37 ± 4.46 (Table 1).

The SLE sample (Table 2) was representative of a population of SLE patients, with a median SLEDAI-2k equal to 4 (0–10), PGA 1 (0–2); 63.6% were in remission and 42.4% had at least one item of damage.

Table 3 shows the attributable burden on worsening QoL in patients with SLE in comparison with the attributable burden due to other chronic diseases. Only multiple sclerosis was found to have an attributable burden worse than that due to SLE (7.0 ± 3.5 vs. 5.37 ± 4.46; F = 5.387, df 1, 230, 231, *p* = 0.021); the attributable burden due to SLE was similar to those of major depressive disorder, Wilson’ disease, carotid atherosclerosis, solid cancer and PTSD and was more impactful on the QoL than the attributable burden of disorders such as Celiac disease (5.37 ± 4.46 vs. 2.4 ± 1.0; F = 22.850, df 1, 89, 90, *p* < 0.001), obsessive compulsive disorder (5.37 ± 4.46 vs. 2.9 ± 6.0; F = 4.388, df 1, 117, 118, *p* = 0.0.38), and specific phobia (5.37 ± 4.46 vs. 0.4 ± 4.9; F = 29.837 df 1, 57, 58, *p* < 0.001).

In the overall study sample, 11 people (34.37%) screened positive from depressive episodes with a PHQ-9 score > 7; their score for SF-12 was 25.81 ± 6.61, against 35.24 ± 7.35 of the 21 people with negative scores for PHQ-9, and the difference was statistically significant (Anova 1 way, DF 1, 30, 31; F = 12.506, *p* < 0.0001). Starting from these data, it can be calculated that the attributable burden to the major depressive episode in worsening the H-Qol in SLE was 9.43 ± 5.10 (Table 4).

## 4. Discussion

The study highlights that depression significantly impacts the QoL of patients with SLE. Specifically, it is observed that SLE itself already has a quality-of-life impact similar to that of many severe chronic disorders. However, what makes SLE unique is the substantial impact of a concurrent depressive episode on the deterioration of quality of life. This influence is particularly evident in cases of solid tumors, while the effect of comorbidity with depressive episodes on the decline in quality of life is less pronounced in other conditions, with multiple sclerosis being the only pathology associated with a greater decline in quality of life compared to SLE.

Overall, this investigation underscores the link between depression and compromised quality of life in SLE patients, emphasizing the importance of prevention, early identification, and treatment of mood disorders in this population.

This study found that SLE has a quality-of-life impact similar to that of many serious chronic disorders such as major depressive disorder, Wilson’s disease, cerebral atherosclerosis, solid tumors, and stress disorder. However, what distinguishes SLE is the significant impact of a concurrent depressive episode on the quality-of-life impairment. This similarity is only observed in cases of solid tumors, whereas the influence of comorbidity with depressive episodes of quality-of-life decline is less pronounced in other disorders, including multiple sclerosis, which is the only condition associated with a greater decline in quality of life compared to SLE.

The term ‘major depressive episode’ and not ‘major depressive disorder’ was used prudently because the diagnosis was conducted with a screening tool that identifies the current episode; on the contrary, the diagnosis of major depressive disorder is a lifetime diagnosis that must exclude previous episodes of mania [37]. The depressive episode is, in fact, common to major depressive disorder and bipolar disorder; a major depressive episode has a frequency clearly higher than that of bipolar disorder (of the order of 5/1) [38]. However, it is correct not to identify the episode with the disorder because many sub-threshold conditions of bipolar disorder frequently have positive results in screening tests that are also due to non-pathological hyperactivity and therefore ‘impact’ on the percentage of positives [39,40]. The study therefore concerns the global impact of mood disorders rather than simple depression.

The observational design of this cross-sectional study does not allow the determination of the causal direction of the associations, such as the relationship between depression and low H-QoL. Specifically, it cannot ascertain whether the impairment linked to depression is a consequence of a shared causal factor, such as brain damage, or if the presence of depression exacerbates the medical condition by demotivating the patient and reducing their resilience to the disorder. Notably, certain findings suggest that in SLE, mood disorders, alongside a decline in cognitive functioning, are correlated with distinct brain damage [41,42].

The conditions of the central nervous system in individuals with SLE and concurrent depression were examined and compared to those with SLE but without depression. Nevertheless, cohort studies involving substantial sample sizes are imperative for reinforcing the evidence regarding the association and for elucidating the causal relationship of this association.

H-QoL appears to be linked with unfavorable outcomes in SLE [43] alongside depressive symptoms and stress [14]. Therefore, the prevention, early identification, and treatment of mood disorders are crucial for individuals with SLE and should consider their multifactorial pathogenesis involving both direct (e.g., antineuronal antibodies, pain) and indirect (e.g., glucocorticoids, relationship issues, comorbidities) factors [44,45]. This not only aids in recognizing the risk factors for compromised H-QoL but also in addressing these factors effectively.

This study has several limitations. The limited sample size reduces the statistical power in the associations between H-QoL and depressive episodes in SLE. Furthermore, it should be acknowledged that the comparison of SF-12 scores was made among different pathologies without accounting for potential determinants specific to each condition that could impact H-QoL. These determinants include variations in gender and age distribution and differences in the frequency of comorbid conditions. However, by conducting the comparison not between distinct pathologies directly, but by contrasting the SF-12 scores for each particular pathology within a sample diagnosed with that pathology and a corresponding control group selected from the same database, balanced for sex and age concerning the specific pathology, thereby mitigating potential biases.

## 5. Conclusions

This study emphasizes that SLE has a substantial impact on individuals’ quality of life, comparable to several severe chronic disorders, including major depressive disorder, Wilson’s disease, cerebral atherosclerosis, solid tumors, and post-traumatic stress disorder. Notably, SLE distinguishes itself through the significant influence of concurrent depressive episodes on the impairment of quality of life. This similarity in the quality-of-life impact is particularly significant in the case of solid tumors, whereas the effect of comorbid depressive episodes on the decline in quality of life is less pronounced in other conditions. Among these, multiple sclerosis is the only condition associated with a more substantial decline in quality of life compared to SLE.

However, it is important to acknowledge that the cross-sectional observational design of this study does not permit the determination of causal relationships. For example, it cannot establish whether the impairment linked to depression is a consequence of a shared causal factor such as brain damage, or if depression exacerbates the medical condition by demotivating patients and reducing their resilience. Notably, some findings suggest that in SLE, mood disorders, combined with cognitive decline, are associated with distinct brain damage.

To further strengthen the evidence regarding these associations and gain a more comprehensive understanding of their causal links, conducting cohort studies with larger sample sizes is imperative.

In conclusion, this study underscores the significance of addressing mood disorders, particularly depression, in individuals with SLE due to the disorders’ substantial impact on health-related quality of life.

The design of this study precludes the determination of causal relationships. It is not possible to establish whether the impairment associated with depression is a consequence of a shared causal factor, such as brain damage, or if depression exacerbates the medical condition by demotivating patients and reducing their resilience.

Furthermore, despite the recognition of the significant impact of depression on health-related quality of life in individuals with SLE, there may be unconsidered confounding factors influencing these results, such as socioeconomic variables or other unmentioned concomitant medical conditions.

The relatively small sample size constitutes a significant limitation. This could affect the generalizability of the results and the representativeness of the SLE population.

To gain a deeper understanding of causal relationships and strengthen the evidence, it is imperative to conduct cohort studies with larger sample sizes. Additional research is necessary to confirm and extend the current findings.

Additionally, these findings acknowledge the study’s limitations, including the relatively small sample size and emphasizes the need to consider potential determinants specific to each condition in future research in order to enhance our understanding of these intricate associations.

## Figures and Tables

**Table 1 healthcare-12-00233-t001:** Study and control samples, matched by sex and age.

SLE Patients		CONTROLS from Community Sample of the Italian General Population	Differences
	N (%)	N (%)	
**Female**	29 (90.62)	116 (90.62)	Perfect matched
Age	44.00 ± 13.64	44.00 ± 13.64	Perfect matched
**SF-12 Mean Score**	32.96 ± 7.09	38.33 ± 6.04	**Anova 1 way DF 1, 53, 54; F = 18.161, *p* < 0.0001**
**Total**	32	128	

**Table 2 healthcare-12-00233-t002:** Clinical and serologic characteristics of the SLE patients.

SLE Features	
**Female**	29 (90.62)
Age	44.00 ± 13.64
Disease duration, median	11.8 (2.8–26.1)
**SLEDAI-2k**	3 (0–10)
**PGA**	1 (0–2)
**SLICC-Damage Index**	0 (0–1)
**Active SLE manifestations**	12 (37.50)
**Cutaneous rash**	6 (18.75)
**Sinovytis**	5 (15.62)
**Hematologic**	2 (6.25)
**Renal**	4 (12.5)
**Neuropsychiatric**	2 (6.25)
**ANA positive**	32 (100)
**Anti-dsDNA**	18 (56.25)
**Anti-Sm**	9 (28.12)
**Anti-Ro/SSA**	15 (46.87)
**Anti-neuronal antibodies**	16 (50)
**Anticardiolipin (IgM and/or IgG)**	6 (18.75)
**Anti-B2glicoprotein 1 (IgM and/or IgG)**	5 (15.62)
**Lupus anticoagulant**	8 (25)
**Conventional MRI findings**	14 (43.75)
**WMHI**	12 (37.50)
**GMHI**	3 (9.37)
Inflammatory-type lesions	0
Areas of resolved infarcts	2 (6.25)
**Brain atrophy**	3 (9.37)
**Antiphoshpilipid Syndrome**	4 (12.5)
**Hypertension**	10 (31.25)
**Diabetes**	2 (6.25)
**Obesity**	3 (9.37)

SLEDAI: Systemic Lupus Erythematosus Disease Activity Index. PGA: Physician Global Assessment. MRI: magnetic resonance imaging. WMHI: white matter hyperintensity. GMHI: grey matter hyperintensity. Unless otherwise specified, numbers are absolute values and numbers in brackets are percentages.

**Table 3 healthcare-12-00233-t003:** Attributable burden on worsening QoL in patients with SLE in comparison to the attributable burden due to other chronic diseases.

Disease	SF-12 (Mean ± SD)	Attributable Burden on QOL	Comparison with SLE (One-Way ANOVA)
Major Depressive Disorder Carta et al., 2012 [18]	33.8 ± 9.2	5.6 ± 3.6 (N = 37)	F = 0.055, df 1, 66, 67 *p* = 0.815
Multiple Sclerosis (Carta et al., 2014) [26]	29.5 ± 7.3	7.0 ± 3.5 (N = 201)	F = 5.387, df 1, 230, 231 *p* = 0.021
Wilson’ Disease (Carta, Mura et al., 2012) [30]	33.8 ± 9.0	4.4 ± 1.7 (N = 23)	F = 0.9787, df 1, 52, 53 *p* = 0.327
Carotid atherosclerosis (Carta, Lecca et al., 2015) [29]	30.6 ± 8.1	6.2 ± 5.0 (N = 46)	F = 0.538, df 1, 71, 72 *p* = 0.466
Celiac Disease (Carta et al., 2015) [27]	35.83 ± 5.72	2.4 ± 1.0 (N = 60)	F = 22.850, df 1, 89, 90 *p* < 0.001
Obsessive Compulsive Disorder (Carta et al., 2018) [32]	35.4 ± 6.9	2.9 ± 6.0 (N = 88)	F = 4.388, df 1, 117, 118 *p* = 0.0.38
PTSD (Sancassini et al., 2019) [33]	36.3 ± 6.1	3.9 ± 1.0 (N = 26)	F = 2.703, df 1, 55, 56 *p* = 0.106
Specific Phobia (Sancassiani, Romano et al., 2019) [34]	38.3 ± 5.2	0.4 ± 4.9 (N = 28)	F = 29.837 df 1, 57, 58 *p* < 0.001
Solid cancer (Aviles Gonzales et al. 2021) [35]	32.34 ± 6.764	4.67 ± 6.64 (N = 151)	F = 315; df 1, 180, 181 *p* = 0.576
Sistemic Lupus Erythematosus	32.96 ± 7.09	5.37 ± 4.46 (N = 32)	

**Table 4 healthcare-12-00233-t004:** Attributable burden on worsening QoL due to Major depressive episodes in people with SLE, in comparison to that due to major depressive episodes in other chronic diseases.

-	Attributable Burden to Major Depressive Disorder	One-Way ANOVA F (df)	*p*
Solid Cancers (N = 151) (Aviles-Gonzales et al., 2021) [35]	10.1 ± 5.7	0.368 (1, 180, 181)	0.545
Multiple Sclerosis (N = 201) (Carta et al., 2014) [26]	2.9 ± 7.4	22.451 (1, 230, 231)	<0.0001
Fibromyalgia (N = 71) (Sancassiani et al., 2017) [36]	4.77 ± 5.76	15.102 (1, 100, 101)	<0.0001
Wilson’s Disease (N = 61) (Carta, Mura et al., 2012) [30]	3.2 ± 7.9	15.868 (1, 90, 91)	<0.0001
Celiac Disease (N = 60) (Carta et al., 2015) [27]	3.4 ± 5.4	26.450 (1, 89, 90)	<0.0001
Carotid Atherosclerosis (N = 46) (Carta, Lecca et al., 2015) [29]	3.4 ± 8.2	13.269 (1, 75, 76)	<0.0001
Sistemic Lupus Erythematosus (N = 31)	9.43 ± 5.10	Pivot	

## Data Availability

All data generated or analyzed during this study are included in this published article.

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
