# Peer review of "Quality of Life in Systemic Lupus Erythematosus and Other Chronic Diseases: Highlighting the Amplified Impact of Depressive Episodes"

_healthcare, 2024, doi:10.3390/healthcare12020233_

Round 1

Reviewer 1 Report

Comments and Suggestions for Authors

 I believe that research on SLE, a chronic disease that seriously affects the quality of life of subjects, is important. Thank you for the opportunity to read your research. I would like to offer a few humble opinions.

1. I suggest revising the study title to be concise.

2. In the introduction, discussing the relationship between SLE and H-QoL, it is mentioned that insufficient research has been done on this. However, it is necessary to specifically mention why this study was not sufficient and what its limitations were. 

3. It does not seem appropriate to explain SLE by comparing it to skin diseases. It is more appropriate to explain it by comparing it with systemic disease or autoimmune disease.

4. In the last part of the introduction, the purpose of this study is somewhat ambiguous. It is suggested to write additional sentences.

5. Case-control studies have the advantage of being able to cost-effectively consider various factors compared to community subjects when large-scale cohort studies are difficult. However, recall bias or selection bias may occur, and it may be difficult to clearly explain the causal relationship of the results. Please describe this in more detail in the study sample section.

6. Many sentences in the discussion were written as an introduction and there is a lack of comparative discussion with previous studies. Most of the discussions currently written are about the limitations of the study and its results. Please compare and discuss the significant impact of major depression symptoms in SLE patients on the H-QoL of SLE patients.

Author Response

  1. I suggest revising the study title to be concise.

Thank you for your comment. We have modified the article title, emphasizing the most prominent aspects of the study.

  1. In the introduction, discussing the relationship between SLE and H-QoL, it is mentioned that insufficient research has been done on this. However, it is necessary to specifically mention why this study was not sufficient and what its limitations were. 

We have included the answer to this question within the conclusion paragraph.

  1. It does not seem appropriate to explain SLE by comparing it to skin diseases. It is more appropriate to explain it by comparing it with systemic disease or autoimmune disease.

Corrected

  1. In the last part of the introduction, the purpose of this study is somewhat ambiguous. It is suggested to write additional sentences.

We have modified the introduction to make the purpose of our study clearer.

  1. Case-control studies have the advantage of being able to cost-effectively consider various factors compared to community subjects when large-scale cohort studies are difficult. However, recall bias or selection bias may occur, and it may be difficult to clearly explain the causal relationship of the results. Please describe this in more detail in the study sample section.

We have highlighted this aspect in the study sample section 

  1. Many sentences in the discussion were written as an introduction and there is a lack of comparative discussion with previous studies. Most of the discussions currently written are about the limitations of the study and its results. Please compare and discuss the significant impact of major depression symptoms in SLE patients on the H-QoL of SLE patients.

Thank you for ypur comment. We have modified the discussion.

Reviewer 2 Report

Comments and Suggestions for Authors

I would like to thank the authors for their great work and effort.

Author Response

I would like to thank the authors for their great work and effort. I have the following comments:

Very long title: I suggest highlighting the new outcome: The depressive impact of SLE in comparison to other chronic illness.

Thank you for your comment. We have modified the article title, emphasizing the most prominent aspects of the study.

Introduction is very nice and comprehensive.

Material and methods: very clearly explained

I am not sure I can get how did you compare the attributable burden of SLE to other diseases. Why did not you choose instead of the healthy control, patients with different diseases? And what was the aim of recruiting healthy control inspite it is well established that SLE affects the QoL?

Our study originated from the assessment of Systemic Lupus Erythematosus on the reduction of Health-Related Quality of Life, calculated as the difference between the SF-12 scores of a sex- and age-matched community control group and the study cohort. Subsequently, the impact of SLE on worsening HRQoL was compared with the burden associated with other diseases, as determined in previous case-control studies that utilized the same dataset to select control groups. Finally, the burden directly linked to depression was computed as the average deviation in SF-12 scores between individuals with SLE. A similar approach was employed in previous case-control studies that utilized the same dataset to form control groups. Thus, we compared the burden attributed to experiencing a Major Depressive Episode with that observed in the presence of a Major Depressive Episode in other chronic conditions

  • 5- Where the SLE patients with major depressive symptoms suffering from lupus cerebritis or no?

Thank you for your comment. We add patients description and table 3 in our manuscript.

  1. References are upto date.

Reviewer 3 Report

Comments and Suggestions for Authors

In modern society, depression has become one of the most common and serious diseases. It has been reported that depression in SLE was associated with both disease-related factors (increased disease activity, musculoskeletal and skin system involvement) and modifiable factors (lower education, financial strain, physical inactivity).

However, we do not know what factors could have played a decisive and important role, since the characteristics of SLE patients were not presented in the manuscript.

The main characteristics of the disease are presented in paragraph 2.2 “Study sample”, however, there is no clinical and laboratory characteristics of patients with SLE.

It is unclear whether there was CNS damage in the SLE patients included in the study, especially since the authors repeatedly discuss the effect of various brain damages on depression.

It is advisable to provide a definition of a depressive episode in the manuscript.

It is not clear why the authors included in the control group patients with a more severe rheumatic disease - systemic sclerosis - along with chronic non-rheumatic diseases. Systemic sclerosis, which is known to be more severe, may have significantly altered the health-related quality of life (H-QoL) of the control group.

It is necessary to change and shorten the conclusion, leaving only your own data and removing literature data.

There are some minor comments.

1.      The abbreviation SLE is duplicated three times (lines 37, 185 and 206), the abbreviation H-Qol – twice (lines 31 and 59-60).

2.      No transcript “Qol - quality of 13 life” (line 23, 24, 122, 131,139).

3.      No transcript the disease “PTSD - post-traumatic stress disorder” (line 127 and Table 2).

4.      Terms are written incorrectly: rheumatological diseases (rheumatological diseases), line 39; Carotidal atherosclerosis (Carotid atherosclerosis), line 126.

5.      It is necessary to provide a list of diseases in the control group in text of paragraph 2.2. “Study Sample”.

6.      The tables are not logically distributed throughout the text. You cannot start a paragraph 3 “Results” with a table 1.

7.      In Table 1 it would be more clear to present data on SLE, as in Table 3.

8.      One sentence repeated twice – lines 176-178 and 183-185.

9.       In the “Discussion” paragraph, when listing chronic diseases, those that were not listed in the manuscript are listed (lines 153-154).

Comments on the Quality of English Language

Moderate editing of English language required

Author Response

In modern society, depression has become one of the most common and serious diseases. It has been reported that depression in SLE was associated with both disease-related factors (increased disease activity, musculoskeletal and skin system involvement) and modifiable factors (lower education, financial strain, physical inactivity). However, we do not know what factors could have played a decisive and important role, since the characteristics of SLE patients were not presented in the manuscript. The main characteristics of the disease are presented in paragraph 2.2 “Study sample”, however, there is no clinical and laboratory characteristics of patients with SLE. It is unclear whether there was CNS damage in the SLE patients included in the study, especially since the authors repeatedly discuss the effect of various brain damages on depression. It is advisable to provide a definition of a depressive episode in the manuscript. It is not clear why the authors included in the control group patients with a more severe rheumatic disease - systemic sclerosis - along with chronic non-rheumatic diseases. Systemic sclerosis, which is known to be more severe, may have significantly altered the health-related quality of life (H-QoL) of the control group. It is necessary to change and shorten the conclusion, leaving only your own data and removing literature data.

Answer:

I thank the reviewer for the comments. It is well established in the literature that depression significantly impacts the quality of life, especially in individuals with serious conditions such as systemic lupus erythematosus (SLE). The aim of our study was to assess how SLE, by itself, has a substantial impact on the quality of life and how a concurrent depressive episode may further compromise it. Our study highlighted the association of SLE with a concurrent depressive episode affecting the compromise of the quality of life. Therefore, our study does not allow determining the causal direction of the associations, such as the relationship between depression and low quality of life. We cannot establish whether the deterioration associated with depression is a consequence of a shared causal factor, such as brain damage, or if the presence of depression worsens the medical condition by demotivating the patient and reducing their resilience to the disorder. There is a clear need to conduct studies with substantial sample sizes to strengthen the evidence and clarify the causal relationship of this association. We have added Table 2 and some comments in the Results chapter clinical with laboratory characteristics of patients with Systemic Lupus Erythematosus 

There are some minor comments.

  1. The abbreviation SLE is duplicated three times (lines 37, 185 and 206), the abbreviation H-Qol – twice (lines 31 and 59-60).

ok

  1. No transcript “Qol - quality of 13 life” (line 23, 24, 122, 131,139).

ok

  1. No transcript the disease “PTSD - post-traumatic stress disorder” (line 127 and Table 2).

ok

  1. Terms are written incorrectly: rheumatological diseases (rheumatological diseases), line 39; Carotidal atherosclerosis (Carotid atherosclerosis), line 126.

       ok

  1. It is necessary to provide a list of diseases in the control group in text of paragraph 2.2. “Study Sample”.

       Added in study sample.

  1. The tables are not logically distributed throughout the text. You cannot start a paragraph 3 “Results” with a table 1 Answer: We have modified in the Results section. 
  2. In Table 1 it would be more clear to present data on SLE, as in Table 3. Answer: ok
  3. One sentence repeated twice – lines 176-178 and 183-185 Answer: ok
  4. In the “Discussion” paragraph, when listing chronic diseases, those that were not listed in the manuscript are listed (lines 153-154). Answer: ok

Round 2

Reviewer 1 Report

Comments and Suggestions for Authors

After reviewing the revisions, it has been appropriately validated. However, I am curious about the significance of the newly added analysis, Table 2, which appears to depict the characteristics of the subjects. Without a comparative discussion with other studies, it would be beneficial to reconsider the significance of this information.

Author Response

I thank the reviewer for this important consideration. We wanted to add a descriptive table for the characteristics of subjects affected by Systemic Lupus Erythematosus (SLE). We will incorporate this suggestion for further studies that take into account these currently descriptive data.